# Total Synthesis of 4-*epi*-Bengamide E

**DOI:** 10.3390/molecules29081715

**Published:** 2024-04-10

**Authors:** Gabriella Vitali Forconesi, Andrea Basso, Luca Banfi, Davide Gugliotta, Chiara Lambruschini, Marta Nola, Renata Riva, Valeria Rocca, Lisa Moni

**Affiliations:** Department of Chemistry and Industrial Chemistry, University of Genova, Via Dodecaneso, 31, 16146 Genova, Italy; gabriella.vitali.f@gmail.com (G.V.F.); andrea.basso@unige.it (A.B.); luca.banfi@unige.it (L.B.); davide.gugliotta@outlook.com (D.G.); chiara.lambruschini@unige.it (C.L.); nola.marta@gmail.com (M.N.); renata.riva@unige.it (R.R.); valeria.marisa.rocca@unige.it (V.R.)

**Keywords:** biocatalysis, multicomponent reactions, natural products, total synthesis

## Abstract

Bengamide E is a bioactive natural product that was isolated from Jaspidae sponges by Crews and co-workers in 1989. It displays a wide range of biological activities, including antitumor, antibiotic, and anthelmintic properties. With the aim of investigating the structural feature essential for their activity, several total syntheses of Bengamide E and its analogues have been reported in the literature. Nevertheless, no synthesis of the stereoisomer with modification of its configuration at C-4 carbon has been reported so far. Here, we report the first total synthesis of the 4-*epi*-Bengamide E. Key reactions in the synthesis include a chemoenzimatic desymmetrization of biobased starting materials and a diastereoselective Passerini reaction using a chiral, enantiomerically pure aldehyde, and a lysine-derived novel isocyanide.

## 1. Introduction

Bengamides are a vast family of natural products of marine origin isolated by Crews and coworkers [1,2] in the late 1980s from an undescribed specimen of an orange sponge belonging to the Jaspidae family. Although a long time has passed since the discovery, their unique structures continue to attract huge interest of the synthetic community thanks to their prominent anticancer activity in humans [3,4]. For example, they can act as methionine aminopeptidase (MetAp1 and MetAp2) [5] and nuclear factor κB (NF-κB) [6] inhibitors. More recently, some members of bengamides have been isolated from terrestrial myxobacteria Myxococcus virescens by Crews [6] and Brönstrup [7].

Among the 21 different bengamides isolated from marine sponge and myxobacteria, Bengamide E consists of a carbon chain containing four contiguous stereocenters, a double bond with E configuration, and a secondary amide which binds an aminocaprolactamic unit (Figure 1, top). Due to their interesting pharmacological applications, several total syntheses of Bengamide E and its analogues have been reported in the literature [8,9,10,11,12,13,14,15,16,17,18,19,20,21,22,23,24,25,26,27], and their cytotoxicity has been evaluated, showing the structural features that are essential for their activity. In 2013, Sarabia and coworkers published the total synthesis of 2-*epi* and 2,3-bis-*epi*-Bengamide E [25], while Zhou et al. reported the synthesis of the 3,4-bis-*epi*-Bengamide E [14]. All these stereochemical changes resulted in a complete loss of antitumor activity (Figure 1, bottom). Very recently, the total synthesis of 5-*epi*-Bengamide E has been reported by Perali and coworkers, but the biological evaluation is still under investigation [28]. 

To our knowledge, no synthesis of the stereoisomer of Bengamide E with modification of chirality at C-4 carbon has been reported so far. In this paper, we present a new stereoselective approach to the unprecedent 4-*epi*-Bengamide E. Contrary to the previous syntheses, which were all target-oriented, we tried to develop a highly convergent, multicomponent reaction (MCR)-based approach to this structure, which may allow the easy introduction of diverse sidechains and the exploration of stereochemical diversity. As a first example, we will describe here the synthesis of an unprecedented isomer of natural Bengamide E.

The key step in our synthetic plan is a diastereoselective Passerini reaction using a chiral, enantiomerically pure aldehyde. The Passerini three-component reaction [29] involves an isocyanide, an aldehyde (or a ketone), and a carboxylic acid to produce an α-acyloxyamide (Figure 2). It has recently emerged as a powerful method for the preparation of natural products and APIs [29,30,31,32], thanks to the advantages that multicomponent reactions in general possess, such as atom and step economy and experimental simplicity. 

The main benefits of this strategy are (i) sustainability, thanks to the cooperation of multicomponent reactions with biocatalysis and/or with the use of renewable starting materials derived from biomass; (ii) the flexibility and convergence of the approach, which can allow the preparation of several analogs in a short and efficient manner. 

In recent years, we have been particularly interested in the use of enantiopure chiral building blocks, synthesized using the chemoenzimatic treatment of biobased starting materials, and their use in diastereoselective Passerini reactions [33]. Actually, when aldehydes different from formaldehyde or unsymmetric ketones are used, this reaction generates a new stereogenic center, generally with poor diastereoselectivity. In this context, we have developed the diversity-oriented synthesis of chiral polysubstituted O- and N-heterocycles employing erythritol as the starting material [34,35,36].

Herein, we present a different elaboration of *meso*-erythritol **1**, using it as a valuable biobased building block for the total synthesis of 4-*epi*-Bengamide E **2**. The convergent retrosynthetic strategy is depicted in Figure 3 and stems from two key disconnections through the C1-C2 and C5-C6 bonds of the carbon chain. We envisioned constructing **2** through methylation and subsequent full deprotection of the advanced intermediate **3**, whose synthesis involves a diastereoselective Passerini reaction of chiral aldehyde **4** with chiral isocyanide **5** and acetic acid, followed by acetate hydrolysis. Enantiopure isocyanide **5** can be conveniently obtained starting from chiral pool compounds (L-lysine), while, for aldehyde **4**, a stepwise synthesis involving the chemoenzimatic desymmetrization of *meso*-diol **1** and the reaction with lithium acetylide of 3-methylbutyne was planned.

## 2. Results and Discussion

We first studied the enantioselective preparation of aldehyde **4** starting from *meso*-diol **1**, which can be conveniently obtained employing biobased materials (Figure 4). As reported by us [36] and others [37], *meso*-erythritol can be converted in three straightforward steps in compound **1**. Alternatively, during this project, we found that the desired compound **1** could also be prepared starting from D-isoascorbic acid, the C-5 epimer of L-ascorbic acid, an important renewable materials produced through the microbial process from sugars [38]. Based on a procedure reported in the literature [39], D-isoascorbic acid was oxidized in the presence of hydrogen peroxide and subsequently protected as acetonide, affording lactone **6** high yield (75%), which was then reduced with LiAlH_4_, giving **1** without need of further chromatographic purifications. Considering the operational simplicity, the greenness and cheapness of the starting material and reagents and the higher yield, this approach was found to be more sustainable and convenient. 

For the synthesis of alcohol **8**, bearing the appropriate protecting group, we initially used the enzymatic monoacetylation of diol **1** using vinyl acetate as the solvent, as previously reported by us [36], followed by a Mitsunobu reaction with *p*-methoxyphenol and a subsequent hydrolysis of the ester moiety under basic conditions (Figure 5). The change in the protecting group is necessary, not only to obtain the desired enantiomer, but mainly because of the instability of the acyl group in the subsequent reactions, as demonstrated by previous studies in our lab (other protecting groups such as TBS or TBDPS resulted in instability during the following reduction of the triple bond). 

Thus, after careful evaluation, *p*-methoxyphenyl (PMP) was chosen as the protecting group, since it involves orthogonal cleavage conditions (reaction with ammonium cerium (IV) nitrate) with respect to the other protecting groups planned for the synthesis (acetonide and *t*-butyl-dimethyl-silyl (TBS) group are removed under acidic conditions). Unfortunately, chiral HPLC analysis of **8** revealed a low enantiomeric excess, probably due to a partial migration of the acetyl group when compound **7a** was reacted under the Mitsunobu conditions (see Appendix A). After several unsuccessful attempts to find **8,** avoiding the Mitsunobu reaction (see Appendix A), we successfully solved this problem by performing the enzymatic desymmetrization with the more steric hindered vinyl butyrate. The enzymatic acylation was performed following the procedure reported in the literature [36], allowing us to isolate the monoacylated compound **7b** in excellent yield (98%). A mitsunobu reaction was carried out at room temperature in the presence of *p*-methoxyphenol (PMP-OH), triphenylphosphine, and di-*tert*-butyl azodicarboxylate (TBAD) and the crude residue was directly subjected to the next reaction with potassium hydroxide in MeOH to remove the acyl group. In this way, product **8** was obtained with a high yield (89%), even on a large scale (about 5 g), and the enantiomeric excess (96%) was completely retained. It is worth noting that the enantiomer of **7b** is easily available as well through enzymatic hydrolysis of the corresponding diacylated derivative [36].

Alcohol **8** was then converted to the corresponding aldehyde using Swern oxidation. Initial attempts to directly introduce the olefinic residue, using organolithium or Grignard reagents did not lead to good results, due to the difficult preparation of the needed alkenyl reagents. Therefore, we decided to explore the nucleophilic addition of an acetylide instead (Figure 6). This strategy leads to the formation of a propargyl alcohol, which can be selectively reduced to the *E-*allylic alcohol through treatment with aluminum hydrides, such as LiAlH_4_ or Red-Al^®^ [40,41,42]. This method presents two advantages: acetylides can be easily obtained from the corresponding terminal alkynes through a metalation reaction (lithium–hydrogen exchange), and, furthermore, many alkynes are commercially available, making the synthesis of a diversity-oriented library of analogs easier.

So, the addition of the lithium acetylide generated using treatment of the commercially available 1-methylbutyne with *n-*BuLi in THF, provided a mixture of separable diastereoisomers **9** with an excellent yield, but without any stereochemical induction. Therefore, in order to increase the d.r., we converted the mixture into the corresponding propargyl ketone via Dess–Martin Periodinane (DMP) oxidation followed by diastereoselective reduction with K-selectride at a low temperature to form the corresponding alcohols **9** as a 24:76 separable mixture of diastereomers, in which **9 *anti*** was the major product (55% isolated yield) [43]. The relative configuration of compounds **9** was determined with chemical conversion into the corresponding lactones **10** with treatment with cerium(IV) ammonium nitrate (CAN) to deprotect the PMP group and subsequent TEMPO/BAIB oxidation (Figure 6). Since lactones **10** are rather rigid compounds, the examination of the *J* coupling relationship between H6 and H6a in the ^1^H-NMR spectra allowed us to establish the relative configuration of the C-4 center: the *anti*-isomer shows a *J*_H6,H6a_ around 0 Hz according to a dihedral angle close to 90°, while the *J*_H6,H6a_ for the *syn*-isomer result are ~4 Hz. The detailed NMR studies are described in the Appendix A.

Then, the allyl alcohol **11** was prepared using hydroalumination with sodium bis-(2-methoxyethoxy)-aluminumhydride (RED-Al^®^) on the single isomer **9 *anti*** (Figure 7) [42,44,45]. The preliminary results obtained were not satisfactory, because of the presence of unreacted starting material, ending up with an only moderate yield (67% after 17 h). On the other hand, when the reaction was performed at reflux rather than at room temperature [46], the allylic alcohol **11** was isolated in excellent yield (82%) (Figure 7). Finally, the secondary hydroxy group was protected as *tert*-butyldimethylsilyl (TBS) and the PMP residue was removed in order to obtain the key intermediate **12**, which was a useful precursor for the next step. As previously said, the choice of these two protecting groups was due to their orthogonal behavior. While the protection with TBS afforded the desired product in excellent yield under typical conditions [47,48], the subsequent PMP cleavage result was troublesome. Although the orthogonal PMP group removal in the presence of TBS ethers using CAN has already been reported in the literature [49,50,51], in this case, the acidic environment evidently promotes the concomitant removal of the silyl group. For this reason, the reaction must take place in a very short time, about 10–15 min, and it is extremely important to promptly quench it to avoid the formation of unprotected diol. 

With the key intermediate **12** in hand, we turned our attention to the synthesis of isocyanide **5** (Figure 8). 

Starting from commercially available L-(-)-α-amino-ε-caprolactam **13**, the formamide **14** was prepared through coupling with formic acid, in the presence of dicyclohexylcarbodiimide (DCC) and Et_3_N. Since **14** presents high solubility in water, it must be isolated with filtration on celite to separate diclyclohexylurea, avoiding a difficult aqueous work up, and subsequent column chromatography. Then, the dehydration reaction was carried out to obtain the desired isocyanide. After careful optimization, POCl_3_ was selected as the best dehydrating agent for the reaction, affording product **5** an excellent yield. Although the stereoconservative preparation of chiral α-isocyano amides from the corresponding formamides is less problematic than that of the corresponding esters [52], the use of an organic base such as Et_3_N might lead to some epimerization. For this reason, we have checked the enantiopurity of isocyanide **5** both using chiral HPLC analysis and by employing it in a Passerini reaction with model compounds, demonstrating a substantial retention of e.e. (see Appendix A). 

Since chiral isocyanide **5** has never been reported in the literature and used in a multicomponent reaction, we initially planned to check its reactivity in a Passerini reaction with the enantiopure alcohol **7a** as a simplified model compound. Thus, we investigated the oxidation of **7a** into the corresponding aldehyde **15** and the subsequent Passerini reaction between **15**, **5,** and acetic acid (Table 1). 

Initially, following our previous work on *meso*-erythritol derivatives [34], we used a one-pot protocol, in which **7a** was oxidized with catalytic TEMPO and stoichiometric PhI(OAc)_2_ and subsequently treated with isocyanide **5**. In this one-pot process, the acetic acid is generated in situ as a by-product of PhI(OAc)_2_ decomposition. Isocyanide **5** proved to be poorly reactive, affording product **16** in moderate yield, even with good diastereoselectivity, with **16 *anti*** prevailing (entry 1, Table 1). Furthermore, a certain amount of formamide **14** has been isolated, demonstrating the tendency of **5** to hydrate.

Based on our previous experience with a diastereoselective Passerini reaction employing biobased-derived aldehydes [34,35], we repeated the reaction in the presence of a substechiometric amount of ZnBr_2_, which proved to be very efficient in increasing d.r. in similar reactions (entry 2, Table 1). Unfortunately, in this case, the addition of the Lewis acid led to complete degradation of the substrates. Moreover, we noted the formation of the side-products **17**, where the carboxylic acid originated by the overoxidation of aldehyde **15**, reacted instead of acetic acid in the multicomponent reaction.The easy overoxidation of such *meso*-erythritol-derived compounds strongly affects the yield and makes the purification of the desired products extremely difficult. 

For these reasons, we decided to perform the oxidation of **7a** under Swern conditions (entry 3, Table 1), which completely suppresses overoxidation. After a work-up under slightly acidic conditions, aldehyde **15** was used as such, avoiding chromatography, because of its known instability over silica gel. When **15**, **5,** and acetic acid were submitted to a traditional Passerini reaction, we isolated **16** with a good (69%) overall yield and diastereomeric ratio (*syn*/*anti* 20:80).

With the aim to improve the d.r., we tried the modified Passerini reaction using zinc dicarboxylates. Employing Zn(OAc)_2_, the yield diminished considerably without any improvement in d.r. (entry 4, Table 1).

In view of these results, we selected the two-step protocol involving Swern oxidation for further studies with aldehyde **4** (Table 2).

Since the poor reactivity of **5** and its tendency to rehydrate during the long reaction time, we initially carried out the Passerini reaction with the consecutive addition of small aliquots of **5** (entry 1, Table 2), or using slow addition through a syringe pump (entry 2, Table 2). In both cases, the yields were not satisfactory, even if excellent d.r.ss were detected. However, the careful HPLC-UV analysis of the products obtained revealed the presence of a mixture of so-called ‘truncated Passerini products’ **3**, although they were formed in small amounts (5–10%). Since compounds **3** are the desired products of the next step, their formation should not be a problem. Nevertheless, the presence of **3** posed major difficulty in the analysis of the reaction outcome and purification of the products, making the d.r. values erratic. This suggested we should perform Swern oxidation, a Passerini reaction, and deacetylation in a sequential process, isolating directly deacetylated Passerini products **3**. In this way, the d.r.ss were determined through HPLC-UV analysis of compounds **3** after the tree steps (entries 3–8, Table 2). A series of solvent screening experiments was then performed. While the solvent seems not to affect the diastereoselection (entries 3–5, Table 2), a remarkable increase in the yield was obtained using *i*Pr_2_O (entry 4) instead of CH_2_Cl_2_ (entry 3) or THF (entry 5). These good results prompted us to investigate the effect of the Lewis acid additive, but again without any advantage (entries 6–8, Table 2). In conclusion, key intermediate **3** was obtained with a satisfactory yield over three steps (72%) and good stereoselectivity, employing iPr_2_O as a solvent in the Passerini reaction and without any additive.

The selective O-methylation of **3 *anti*** was particularly troublesome, due to the concomitant N-alkylation of the caprolactam unit. Careful optimization needed to be carried out, and the best conditions turned out to be MeI as the alkylating agent and a slight excess of NaH as the base at −10 °C in THF. Under these conditions, we were able to isolate the desired compound **19** in a 66% yield, together with **20**, where the O- and N-methylations had occurred (Figure 9). It is noteworthy that the chemical elaboration of compound **20** would lead to the epimer of Bengamide F, another member of the bengamide family.

For the conversion of **19** to 4-*epi*-Bengamide **2**, many acid catalyzing deprotection methods were evaluated. While the use of a mixture of TFA/THF/water [36] resulted in complete decomposition, the stepwise removal of the TBS group with TBAF, and the subsequent reaction with AcOH (70% aqueous solution) [24,26], furnished **2**, even produced a very poor yield (6%). Actually, lactone **21** was isolated as a major compound (Figure 9, box). The formation of this product is probably due to the spatial proximity between the OH in position five and the amide carbonyl group, which favors intramolecular attack, expelling the amino-caprolactam unit. Finally, the aqueous HCl (1 N) and THF (2:1) [53] mixture gave the best results, providing **2** with a 45% isolated yield (Figure 9).

## 3. Materials and Methods

^1^H and ^13^C NMR spectra were recorded with a Varian Mercury 300 (at 300 MHz, and 75 MHz, respectively) or a JEOL 400 (at 400 MHz and 101 MHz, respectively).

Unless otherwise stated, NMR spectra were recorded using residual solvent as the internal standard ^1^H NMR: TMS = 0.00; (CD_3_)_2_SO = 2.50; and ^13^C NMR: CDCl_3_ = 77.16; (CD_3_)_2_SO = 39.52. Data for ^1^H NMR spectra are reported as follows: chemical shift (δ ppm), integration, multiplicity, and coupling constants (Hz). Data for ^13^C NMR spectra are reported in terms of chemical shift (δ ppm). Interpretation of spectra has been made also with the aid of gCOSY, gHSQC, and gHMBC experiments. The following abbreviations are used to indicate the multiplicity in NMR spectra: s, singlet; d, doublet; t, triplet; q, quartet; m, multiplet.

IR spectra were recorded directly on solid, oil, or foamy samples, with the ATR (attenuated total reflectance) technique, using a FT Perkin Elmer Spectrum 65 spectrophotometer. TLC analyses were carried out on silica gel plates, viewed at UV (ν = 254 nm) and developed with Hanessian stain (dipping into a solution of (NH_4_)4MoO_4_·4H_2_O (21 g) and Ce(SO_4_)_2_·4H_2_O (1 g) in H_2_SO_4_ (31 mL) and H_2_O (469 mL) and warming. Rf values were measured after an elution of 7–9 cm. Chiral HPLC analyses for the determination of enantiomeric excess were performed on a Daicel Chiral Pak AD 250 × 4.6 mm column, at 25–26 °C with a flow of about 0.8 mL/min (UV detection at ν = 220 nm). HPLC-MS analyses were performed on Synergi Hydro RP 150 × 3 mm column, at 30 °C with a flow of 0.5 mL/min (where not otherwise stated). For MS, the ESI+ ionization method was used. HPLC-UV analyses were carried out on a HP-1100 system (Agilent, Santa Clara, CA, USA) equipped with (a) a HYDRO RP column (150 × 3 mm, 4 μ) at 25 °C with flow = 0.5 mL/min and isocratic elution (CH_3_CN/H_2_O 50:50). Detection was conducted with UV at 220 nm; (b) a C6 PHENYLIC RP column (150 × 3 mm, 3 μ) at 25 °C with flow = 0.38 mL/min and gradient H_2_O/CH_3_CN, A = CH_3_CN—B = H_2_O, 0 min B = 70%, 20 min B = 0%. HRMS: samples, provided at 10 mM in DMSO, were diluted at 50 µM with acetonitrile/water 1:1, and analyzed on a UPLC Acquity system coupled to a Synapt G2 QToF mass spectrometer. MS signals were acquired from 50 to 1200 *m*/*z* ESI positive ionization mode. UPLC was carried out with H_2_O–CH_3_CN–HCO_2_H with an Acquity UPLC BEH C18, 1.7 µM, 2.1 × 50 mm column at 45 °C. Column chromatography was performed with the “flash” methodology using 220–400 mesh silica. Melting points were determined with an electrothermal apparatus (Büchi B-535). Petroleum ether (40–60 °C) is abbreviated as PE. All reactions employing dry solvents were carried out under a nitrogen atmosphere. After extractions, the aqueous phases were always re-extracted three times with the appropriate organic solvent, and the organic extracts were always dried over Na_2_SO_4_ and filtered before evaporation to dryness.

**((4R,5S)-5-((4-methoxyphenoxy)methyl)-2,2-dimethyl-1,3-dioxolan-4-yl)methanol 8:** from **7a**: To a solution of **7a** (110 mg, 0.50 mmol), triphenylphosphine (193 mg, 0.73 mmol) and *p*-methoxyphenol (182 mg, 1.50 mmol) in dry CH_2_Cl_2_ (5 mL) at 0 °C was added to *tert*-butyl azodicarboxylate (175 mg, 0.74 mmol). The mixture was stirred at room temperature for 37 h. Then, the reaction mixture was concentrated and filtered through a short column of silica gel with PE/CH_2_Cl_2_/Et_2_O (1:1:1). The residue was directly diluted with MeOH (2 mL) and treated with KOH (0.80 mL, 6 M in MeOH). The reaction mixture was stirred at room temperature for 6 h, then diluted with saturated NH_4_Cl aq, extracted with Et_2_O, dried (Na_2_SO_4_), and concentrated. The crude residue was eluted from a column of silica gel with PE/AcOEt 3:1 to give **8** (152 mg, 93%, e.e. 76%) as a colorless oil. 

From **7b**: To a solution of **7b** (4.71 g, 21.57 mmol), triphenylphosphine (8.49 g, 32.35 mmol) and *p*-methoxyphenol (8.03 g, 64.70 mmol) in dry CH_2_Cl_2_ (215 mL) at 0 °C was added to *tert*-butyl azodicarboxylate (7.45 g, 32.36 mmol). The mixture was stirred at room temperature for 37 h. Then, the reaction mixture was concentrated and filtered through a short column of silica gel with PE/CH_2_Cl_2_/Et_2_O (1:1:1). The residue was directly diluted with MeOH (108 mL) and treated with KOH (32 mL, 1 M in MeOH). The reaction mixture was stirred at room temperature for 6 h, then diluted with saturated NH_4_Cl aq, extracted with Et_2_O, dried (Na_2_SO_4_), and concentrated. The crude residue was eluted from a column of silica gel with PE/AcOEt (from 2:1 to 1:1) to give **8** (5.16 g, 89%, e.e. 95%) as a colorless oil. The enantiomeric excess was determined using HPLC on a chiral stationary phase. Conditions: column Daicel Chiral Pak AD (250 × 4.6 mm); detector DAD (220 nm); flow 0.8 mL min^−1^. Isocratic elution with *n*-hexane/isopropanol 90: 10. Temperature: 25 °C. Rt 14.3 min. (4R,5S) and 17.0 min (4S,5R). R_f_ = 0.27 (PE/AcOEt 2:1); [α]_D_^20^ = +8.2 (*c* 1.0, CHCl_3_); m.p. 52.3–54.2 °C (CH_2_Cl_2_); IR (ATR): ν = 3519, 3058, 2988, 2938, 2887, 2836, 1509, 1460, 1374, 1335, 1289, 1216, 1182, 1164, 1111, 1089, 1050, 1035, 996, 907, 845, 830, 818, 803, 751, 715, 650 cm^−1^; ^1^H NMR (CDCl_3_, 400 MHz): *δ* = 6.91–6.78 (m, 4H, aromatic H), 4.55 (q, *J* = 6.3 Hz, 1H, CH-CH_2_OPMP), 4.40 (q, *J* = 6.3 Hz, 1H, CH-CH_2_OH), 4.08–4.01 (m, 2H, CH_2_OPMP), 3.88–3.74 (m, 2H, CH_2_OH), 3.77 (s, 3H, OCH_3_), 2.22 (q, *J* = 6.4 Hz, 1H, OH), 1.50 (s, 3H, CH_3_ of acetonide), 1.41 (s, 3H, CH_3_ of acetonide); ^13^C NMR (CDCl_3_, 101 MHz): *δ* = 154.5 (Cq Ar), 152.3 (Cq Ar), 115.7 (2 CH Ar), 114.8 (2 CH Ar), 109.0 (Cq acetonide), 77.4 (CH-CH_2_OH), 75.0 (CH-CH_2_OPMP), 67.2 (CH_2_OPMP), 61.1 (CH_2_OH), 55.8 (OCH_3_), 27.9 (CH_3_ acetonide), 25.3 (CH_3_ acetonide); HRMS (ESI+) *m*/*z*: [M + Na]^+^ Calcd for C_14_H_20_NaO_5_^+^: 291.1203; Found: 291.1106.

**1-((4R,5S)-5-((4-methoxyphenoxy)methyl)-2,2-dimethyl-1,3-dioxolan-4-yl)-4-methylpent-2-yn-1-ol (9):** To a solution of DMSO (1.1 mL, 15.08 mmol), in dry CH_2_Cl_2_ (36 mL), at −70 °C, under a nitrogen atmosphere, a solution of oxalyl chloride in dry CH_2_Cl_2_ (1.43 M, 9.8 mL) was added. The solution was stirred for approximately 10 min, until effervescence ceased. A solution of **8** (1.50 g, 5.59 mmol) in dry CH_2_Cl_2_ (20 + 10 + 6 mL) was added dropwise, and the solution was stirred for 10 min at −70 °C. NEt_3_ (4.3 mL, 30.73 mmol) was then added, and the solution was stirred for 2 h at −70 °C. After this time, the reaction mixture was poured into a mixture of 5% aq (NH_4_)H_2_PO_4_ (90 mL) and 1 M HCl (10 mL) (final pH 4) and extracted with Et_2_O (100 + 30 mL). The organic layer was washed with brine (20 mL), dried (Na_2_SO_4_), and concentrated. The resulting crude aldehyde was rapidly solubilized in THF (20 mL) under Ar and used as such for the next reaction. To a solution of 2,2′-bipyridine (catalytic amount) in dry THF (30 mL) under Ar, at −50 °C, *n-*BuLi (11 mL, 1.6 M in hexane) was added, until a deep red color persisted. Then, 3-methyl-1-butyne (2 mL, 19.55 mmol) was added, and the mixture was stirred for 25 min. After this time, the temperature was kept at −70 °C, and the solution of aldehyde was slowly added to the mixture. The reaction was stirred for 1 h at −50 °C and overnight at room temperature. The reaction was diluted with saturated NH_4_Cl aq., extracted with AcOEt, dried (Na_2_SO_4_) and concentrated. The residue was eluted from a column of silica gel with PE/Et_2_O 3:1 to give **9** (1.64 g, 88%) as a 57:43 mixture of diastereoisomers (PHENYLIC RP column 150 × 3 mm, 3 μm, temp 25 °C, flow = 0.38 mL/min, mobile phase H_2_O/CH_3_CN, A = CH_3_CN—B = H_2_O, 0 min B = 90%, 30 min B = 0%. *R_t_* (*syn*) = 12.0 min, *R_t_* (anti) = 12.4 min).

**Oxidation and diastereoselective reduction to give 9 *anti*:** To a solution of **9** (969 mg, 2.90 mmol), in dry CH_2_Cl_2_ (22 mL), under a nitrogen atmosphere, Dess Martin periodinane (1.35 g, 3.19 mmol) was added at 0 °C, and the reaction was stirred at room temperature for 4.5 h. The mixture was quenched with NaHCO_3_ (5% *w*/*v* aqueous solution)/Na_2_S_2_O_3_ (0.4 M in water) (1:1), extracted with CH_2_Cl_2_, dried (Na_2_SO_4_), and concentrated to afford the corresponding ketone, which was used as such for the next reaction. To a solution of ketone in dry THF (29 mL) at −78 °C under a nitrogen atmosphere, K-Selectride (1 M in THF, 2.9 mL) was added. After stirring at room temperature for 5 h, the reaction was diluted with saturated NH_4_Cl (saturated aqueous solution), extracted with AcOEt, washed with brine, dried (Na_2_SO_4_), and concentrated. The residue was eluted from a column of silica gel with PE/AcOEt 3:1 to give first **9 *anti*** (529 mg, 55%) as a white solid and **9 *syn*** (168 mg, 18%) as a pale-yellow oil. The diastereoisomeric ratio (76:24) was determined on the crude after the reduction with HPLC (PHENYLIC RP column 150 × 3 mm, 3 μm, temp 25 °C, flow = 0.38 mL/min, mobile phase H_2_O/CH_3_CN, A = CH_3_CN—B = H_2_O, 0 min B = 90%, 30 min B = 0%. *R_t_* (*syn*) = 12.0 min, *R_t_* (*anti*) = 12.4 min).

**9 *anti*:** R_f_ = 0.73 (PE/AcOEt 6:4); [α] _D_^25^ = −13.0 (*c* 1.0, CHCl_3_); m.p. 85.8–88.3 °C (CHCl_3_); IR (ATR): ν = 3455, 3222, 2970, 2934, 2835, 1507, 1458, 1381, 1319, 1289, 1228, 1214, 1167, 1125, 1106, 1082, 1038, 884, 856, 824, 727, 639 cm^−1^; ^1^H NMR (CDCl_3_, 300 MHz): *δ* = 6.93–6.86 (m, 2H, 2 CH Ar), 6.86–6.79 (m, 2H, 2 CH Ar), 4.65–4.55 (m, 2H, CH-4 and CHOH), 4.36 (dd, *J* = 10.2, 5.2 Hz, 1H, 1 H of CH_2_), 4.31 (dd, *J* = 6.5, 5.1 Hz, 1H, CH-5), 4.20 (dd, *J* = 10.1, 6.2 Hz, 1H, 1 H of CH_2_), 3.77 (s, 3H, OCH_3_), 2.70 (d, *J* = 5.7 Hz, 1H, OH), 2.57 (pd, *J* = 6.9, 1.8 Hz, 1H, CH of *i*Pr), 1.55 (s, 3H, CH_3_ of acetonide), 1.42 (s, 3H, CH_3_ of acetonide), 1.14 (d, *J* = 6.9 Hz, 6H, 2 CH_3_ of *i*Pr); ^13^C NMR (CDCl_3_, 75 MHz): δ = 154.1 (Cq Ar), 152.4 (Cq Ar), 115.6 (2 CH Ar), 114.5 (2 CH Ar), 109.1 (Cq acetonide), 92.6 (Cq alkyne), 79.3 (CH-5), 77.2 (Cq alkyne), 75.5 (CH-4), 67.3 (CH_2_), 61.7 (CHOH), 55.6 (OCH_3_), 27.3 (CH_3_ acetonide), 25.2 (CH_3_ acetonide), 22.6 (2 CH_3_ of *i*Pr), 20.4 (CH of *i*Pr); HRMS (ESI+) *m*/*z*: [M + Na]^+^ Calcd for C_19_H_26_NaO_5_^+^: 357.1672; Found: 357.1670.

**9 *syn*:** R_f_ = 0.65 (PE/AcOEt 6:4); [α] _D_^20^= −76.3 (*c* 1.0, CHCl_3_); IR (ATR): ν = 3455, 3222, 2970, 2934, 2835, 1507, 1458, 1381, 1319, 1289, 1228, 1214, 1167, 1125, 1106, 1082, 1038, 884, 856, 824, 727, 639 cm^−1^; ^1^H NMR (CDCl_3_, 300 MHz): *δ* = 6.92–6.79 (m, 4H. 4 CH Ar), 4.57 (td, *J* = 6.5, 4.4 Hz, 1H, CH-5), 4.53–4.48 (m, 1H, CH-OH), 4.30 (dd, *J* = 10.1, 4.4 Hz, 1H, 1 H of CH_2_), 4.26 (dd, *J* = 7.3, 6.3 Hz, 1H, CH-4), 4.10 (dd, *J* = 10.0, 6.6 Hz, 1H, 1 H of CH_2_), 3.77 (s, 3H, OCH_3_), 2.55 (pd, *J* = 7.4, 1.9 Hz, 1H, CH of *i*Pr), 2.53 (bs, 1H, OH), 1.54 (s, 3H, C*H_3_* of acetonide), 1.44 (s, 3H, C*H_3_* of acetonide), 1.13 (d, *J* = 6.9 Hz, 3H, C*H_3_* of acetonide), 1.13 (d, *J* = 6.9 Hz, 3H, C*H_3_* of acetonide); ^13^C NMR (CDCl_3_, 75 MHz): δ = 154.2 (Cq Ar), 152.7 (Cq Ar), 115.7 (2 CH Ar), 114.7 (2 CH Ar), 109.6 (Cq acetonide), 93.0 (Cq alkyne), 79.8 (CH-5), 76.8 (Cq alkyne), 75.6 (CH-4), 67.2 (CH_2_), 61.3 (CHOH), 55.8 (OCH_3_), 27.8 (CH_3_ acetonide), 25.4 (CH_3_ acetonide), 22.8 (2 CH_3_ *i*Pr), 20.6 (CH *i*Pr); HRMS (ESI+) *m*/*z*: [M + Na]^+^ Calcd for C_19_H_26_NaO_5_^+^: 357.1672; Found: 357.1670.

**(3aR,6R,6aR)-2,2-dimethyl-6-(3-methylbut-1-yn-1-yl)dihydrofuro[3,4-d][1,3]dioxol-4(3aH)-one (10 *anti*):** To a solution of **9 *anti*** (30 mg, 0.09 mmol), in CH_3_CN (1.5 mL) and H_2_O (450 µL), at 0 °C, CAN (123 mg, 0.22 mmol) was added. After 10 min, the reaction mixture was diluted with saturated aqueous NaHCO_3_ and extracted with CH_2_Cl_2_. The combined organic layers were dried (Na_2_SO_4_) and concentrated to afford the crude diol, which was directly used for the next reaction. To a solution of crude diol in CH_2_Cl_2_ (1 mL), under a nitrogen atmosphere, TEMPO (5 mg, 0.03 mmol) and BAIB (155 mg, 0.48 mmol) were added. After stirring for 4 h at room temperature, the reaction mixture was diluted with CH_2_Cl_2_, washed with Na_2_S_2_O_3_ (0.4 M in water), dried (Na_2_SO_4_), and concentrated. The residue was purified with chromatography using PE/AcOEt 8:1 to create **10 *anti*** (15 mg, 74%) as a pale-yellow oil. R_f_ = 0.86 (PE/AcOEt = 3:2); [α] _D_^20^ = +50.4 (*c* 2.2, CHCl_3_); ^1^H NMR (CDCl_3_, 300 MHz): δ 5.14 (dt, *J* = 2.0, 0.5 Hz, 1H, propargylic CH), 4.86 (d, *J* = 5.2 Hz, 1H, CH-4), 4.75 (d, *J* = 5.3 Hz, 1H, CH-3), 2.59 (heptd, *J* = 6.9, 2.0 Hz, 1H, CH of *i*Pr), 1.47 (s, 3H, CH_3_ of acetonide), 1.39 (s, 3H, CH_3_ of acetonide), 1.17 (d, *J* = 6.9 Hz, 6H, 2 CH_3_ of CH(CH_3_)_2_); ^13^C NMR (CDCl_3_, 75 MHz): δ = 173.7 (Cq of lactone), 114.6 (Cq of acetonide), 96.8 (Cq alkyne), 80.9 (CH-3), 75.0 (CH-4), 73.0 (Cq alkyne), 71.8 (propargylic CH), 26.9 (CH_3_ acetonide), 26.1 (CH_3_ acetonide), 22.5 (2 CH_3_ of *i*Pr), 20.6 (CH of *i*Pr); HRMS (ESI+) *m*/*z*: [M + Na]^+^ Calcd for C_12_H_16_NaO_4_^+^: 247.0941; Found: 247.0940.

**(3aR,6S,6aR)-2,2-dimethyl-6-(3-methylbut-1-yn-1-yl)dihydrofuro[3,4-d][1,3]dioxol-4(3aH)-one (10 *syn*):** Compound **10 *syn*** was obtained as a white foam (18 mg, 50%), starting from **9 *syn***, using the same procedure as above described for **10 *anti***. R_f_ = 0.70 (PE/AcOEt = 3:2); [α] _D_^20^ = +87.9 (*c* 0.8, CHCl_3_); ^1^H NMR (CDCl_3_, 300 MHz): δ 5.17 (dd, *J* = 3.6, 1.8 Hz, 1H, propargylic CH), 4.81 (dd, *J* = 5.6, 3.6 Hz, 1H, CH-4), 4.78 (d, *J* = 5.6 Hz, 1H, CH-3), 2.67 (heptd, *J* = 6.9, 1.9 Hz, 1H, CH of *i*Pr), 1.51 (s, 3H, CH_3_ of acetonide), 1.44 (s, 3H, CH_3_ of acetonide), 1.21 (dd, *J* = 6.9, 1.0 Hz, 6H, 2 CH_3_ of *i*Pr); ^13^C NMR (CDCl_3_, 75 MHz): δ = 173.1 (Cq lactone), 114.7 (Cq acetonide), 97.4 (Cq alkyne), 76.9 (CH-4), 75.7 (CH-3), 71.1 (propargylic CH), 70.4 (Cq alkyne), 26.9 (CH_3_ acetonide), 26.3 (CH_3_ acetonide), 22.6 (2 CH_3_ of *i*Pr), 20.8 (CH of *i*Pr); HRMS (ESI+) *m*/*z*: [M + Na]^+^ Calcd for C_12_H_16_NaO_4_^+^: 247.0941; Found: 247.0940.

**(R,E)-1-((4R,5S)-5-((4-methoxyphenoxy)methyl)-2,2-dimethyl-1,3-dioxolan-4-yl)-4-methylpent-2-en-1-ol (11):** To a solution of **9 *anti*** (556 mg, 1.66 mmol), in dry THF (17 mL), under an argon atmosphere, Red-Al^®^ (3.5 M in toluene, 1.2 mL) was added dropwise at 0 °C and the reaction was stirred under reflux for 4 h. Then, it was cooled to 0 °C and carefully quenched with 1:1 Rochelle salt (30% aqueous solution) and saturated NH_4_Cl aqueous solution. The mixture was stirred for 1 h and then extracted with AcOEt. The organic phase was washed with brine, dried (Na_2_SO_4_), and concentrated. The crude residue was purified using silica gel column chromatography (PE/Et_2_O 3:1) to afford **11** (451 mg, 82%) as a white solid. R_f_ = 0.55 (PE/AcOEt 4:1); [α]_D_^25^ = +4.9 (*c* 1.2, CHCl_3_). m.p. 44.6–47.1 °C; IR (ATR): ν = 3487, 2990, 2957, 2939, 2883, 2867, 2837, 1858, 1670, 1624, 1591, 1506, 1458, 1441, 1412, 1379, 1367, 1329, 1302, 1290, 1250, 1220, 1183, 1167, 1137, 1113, 1081, 1039, 1013, 971, 958, 936, 923, 906, 861, 822, 799, 778, 721, 669, 659, 642, 605 cm^−1^; ^1^H NMR (CDCl_3_, 300 MHz): *δ* = 6.93–6.79 (m, 4H, 4 CH Ar), 5.79 (ddd, *J* = 15.6, 6.5, 1.2 Hz, 1H, *i*Pr-CH=), 5.57 (ddd, *J* = 15.6, 5.7, 1.3 Hz, 1H, *i*Pr-CH=CH), 4.55 (dt, *J* = 6.7, 5.6 Hz, 1H, CH-CH_2_OPMP), 4.38–4.25 (m, 1H, CHOH), 4.19 (dd, *J* = 9.8, 6.8 Hz, 1H, 1 H of CH_2_), 4.13 (dd, *J* = 7.8, 5.7 Hz, 1H, CH-CHOH), 4.02 (dd, *J* = 9.7, 5.5 Hz, 1H, 1 H of CH_2_), 3.77 (s, 3H, OCH_3_), 2.72 (d, *J* = 3.6 Hz, 1H, OH), 2.33 (hept, *J* = 6.9 Hz, 1H, CH of *i*Pr), 1.47 (s, 3H, CH_3_ of acetonide), 1.39 (s, 3H, CH_3_ of acetonide), 1.01 (dd, *J* = 6.8, 1.1 Hz, 6H, 2 CH_3_ of *i*Pr); ^13^C NMR (CDCl_3_, 75 MHz): *δ* = 154.6 (Cq Ar), 152.2 (Cq Ar), 140.6 (*i*Pr-CH=), 125.9 (*i*Pr-CH=CH), 115.8 (2 CH Ar), 114.8 (2 CH Ar), 109.0 (Cq acetonide), 80.2 (CH-CHOH), 75.7 (CH-CH_2_), 70.3 (CHOH), 67.8 (CH_2_), 55.9 (OCH_3_), 31.0 (CH of *i*Pr), 28.1 (CH_3_ acetonide), 25.5 (CH_3_ acetonide), 22.4 (CH_3_ of *i*Pr), 22.3 (CH_3_ of *i*Pr); HRMS (ESI+) *m*/*z*: [M + Na]^+^ Calcd for C_19_H_28_NaO_5_^+^: 359.1829; Found: 359.1816.

**((4S,5S)-5-((R,E)-1-((tert-butyldimethylsilyl)oxy)-4-methylpent-2-en-1-yl)-2,2-dimethyl-1,3-dioxolan-4-yl)methanol (12):** A solution of **11** (381 mg, 1.13 mmol), in dry CH_2_Cl_2_ (6 mL), under a nitrogen atmosphere was treated with 2,6-lutidine (527 μL, 4.53 mmol) and TBS-OTf (624 μL, 2.72 mmol) at 0 °C. After stirring at room temperature for 3 h, the reaction was diluted with saturated aqueous NH_4_Cl and extracted with CH_2_Cl_2_. The combined organic layers were dried (Na_2_SO_4_) and concentrated and the crude residue was filtered through a short column of silica gel (PE/Et_2_O 8:1) and the free alcohol obtained (470 mg, 92%) is directly subjected to the next reaction. To a solution of free alcohol (75 mg, 0.17 mmol) in CH_3_CN (3 mL), a solution of CAN in deionized water (1 mL, 0.4 M) was added dropwise at −15 °C. After stirring for 15 min at −15 °C, the mixture was diluted with NaHCO_3_ (5% *w*/*v* aqueous solution)/Na_2_S_2_O_3_ (0.4 M in water) (1:1) and extracted with CH_2_Cl_2_. The combined organic layers were dried (Na_2_SO_4_) and concentrated, and the crude residue was purified with silica gel column chromatography (PE/CH_2_Cl_2_/Et_2_O 4:1:0.5) to afford **12** (54 mg, 94%) as a pale-yellow oil. R_f_ = 0.18 (PE/CH_2_Cl_2_/Et_2_O 4:1:0.5); [α]_D_^25^= −17.8 (*c* 1.2, CHCl_3_); ^1^H NMR (CDCl_3_, 300 MHz): *δ* = 5.66 (dd, *J* = 15.3, 6.8 Hz, 1H, *i*Pr-CH=), 5.39 (ddd, *J* = 15.4, 7.2, 1.3 Hz, 1H, *i*Pr-CH=CH), 4.44 (dd, *J* = 7.1, 5.5 Hz, 1H, CH-OTBS), 4.19 (q, *J* = 5.8 Hz, 1H, CH-CH_2_OH), 4.01 (t, *J* = 5.6 Hz, 1H, CH-CHOTBS), 3.75 (ddd, *J* = 11.8, 7.7, 5.8 Hz, 1H, 1 H of CH_2_OH), 3.67 (ddd, *J* = 11.9, 6.2, 5.7 Hz, 1H, 1 H of CH_2_OH), 2.95 (dd, *J* = 7.6, 6.2 Hz, 1H, OH), 2.33 (hept, *J* = 6.7 Hz, 1H, CH of *i*Pr), 1.45 (s, 3H, CH_3_ acetonide), 1.34 (s, 3H, CH_3_ acetonide), 1.00 (dd, *J* = 6.8, 1.8 Hz, 6H, 2 CH_3_ of *i*Pr), 0.90 (s, 9H, 3 CH_3_ of TBS), 0.12 (s, 3H, CH_3_ of TBS), 0.08 (s, 3H, CH_3_ of TBS); ^13^C NMR (CDCl_3_, 75 MHz): δ = 141.4 (*i*Pr-CH=CH), 126.2 (*i*Pr-CH=CH), 108.1 (Cq of acetonide), 79.8 (CH-CHOTBS), 77.8 (CH-CH_2_OH), 72.9 (CHOTBS), 61.7 (CH_2_OH), 30.9 (CH of *i*Pr), 27.9 (CH_3_ acetonide), 26.0 (3 CH_3_ of TBS), 25.8 (CH_3_ acetonide), 22.2 (CH_3_ of *i*Pr), 22.0 (CH_3_ of *i*Pr), 18.3 (Cq of TBS), −3.7 (CH_3_ of TBS), −4.4 (CH_3_ of TBS).; HRMS (ESI+) *m*/*z*: [M+Na]+ Calcd for C_18_H_36_NaO_4_Si^+^: 367.2275; Found: 367.2322.

**(S)-N-(2-oxoazepan-3-yl)formamide (14):** To a solution of L -(−)-α-amino-ε-caprolactam hydrochloride **13** (502 mg, 3.05 mmol) in dry CH_2_Cl_2_ (15 mL), Et_3_N (593 µL, 4.25 mmol), formic acid (183 μL, 4.86 mmol), and DCC (877 mg, 4.25 mmol) were added at 0 °C, and the reaction was stirred at room temperature for 19 h. The mixture was filtered through a pad of celite, washing it with CH_2_Cl_2,_ and the solvent was removed under reduced pressure. The residue was purified using silica gel column chromatography (from AcOEt + 2% MeOH to AcOEt + 10% MeOH) to afford **14** as a white amorphous solid (460 mg, 97%). The optical purity of formamide was checked using chiral HPLC analysis on Daicel Chiralpak AD 250 × 4.6 mm column, after standardization with a racemic sample. Flow 1.0 mL/min; isocratic elution with *n*-hexane/*i*PrOH 90:10; temp.: 25 °C; UV detection at 220 nm. Rt 23.4 min (D) and 27.6 min (L). R_f_ = 0.42 (CH_2_Cl_2_/MeOH 9:1); [α]_D_^20^ = +80.04 (*c* 1.01, CHCl_3_); IR (ATR): ν = 3268, 3089, 2972, 2912, 2866, 2850, 1695, 1628, 1517, 1482, 1437, 1381, 1370, 1361, 1335, 1316, 1292, 1278, 1222, 1212, 1122, 1092, 1057, 1043, 978, 946, 910, 851, 835, 804, 759 cm^−1^; ^1^H NMR (CDCl_3_, 300 MHz): *δ* = 8.20 (s, 1H, CHO), 7.14 (bs, 1H, NH), 6.50 (bs, 1H, NH), 4.61 (dd, *J* = 11.2, 6.1 Hz, 1H, CH), 3.52–3.12 (m, 2H, CH_2_C=O), 2.35–1.96 (m, 2H, 2 H of CH_2_), 1.96–1.72 (m, 2H, 2 H of CH_2_), 1.62–1.34 (m, 2H, 2 H of CH_2_); ^13^C NMR (CDCl_3_, 75 MHz): *δ* = 175.2 (C=O caprolactame), 160.3 (C=O formamide), 51.2 (CH), 42.2 (CH_2_), 31.6 (CH2), 28.9 (CH_2_), 28.0 (CH_2_); HRMS (ESI+) *m*/*z*: [M + Na]^+^ Calcd for C_7_H_12_N_2_NaO_2_^+^: 179.0791: Found: 179.0802.

**(S)-3-isocyanoazepan-2-one (5):** To a solution of **14** (122 mg, 0.78 mmol), in dry CH_2_Cl_2_ (4 mL), Et_3_N (512 μL, 3.67 mmol) and POCl_3_ (179 μL, 1.17 mmol) were added dropwise at −30 °C. After stirring for 90 min at −30 °C, the reaction was diluted with saturated NaHCO_3_ aq, extracted with AcOEt, dried (Na_2_SO_4_), and concentrated. The crude residue was purified using silica gel column chromatography (PE/AcOEt 1:5) to afford **5** as a white amorphous solid (98 mg, 91%). The optical purity of isocyanide was not confirmed in this crude product due to the presence of unresolved peaks in the chromatogram. Therefore, the optical purity was checked on model compounds **16** derived from a Passerini reaction of **5** and **7a** (See Appendix A). R_f_ = 0.63 (CH_2_Cl_2_/MeOH 9:1); [α]_D_^25^= −11.2 (*c* 1.0, CHCl_3_); IR (ATR): ν = 3328, 3223, 3099, 2992, 2948, 2925, 2858, 2148, 1670, 1478, 1466, 1436, 1428, 1359, 1331, 1323, 1291, 1274, 1248, 1111, 1092, 1075, 1038, 1015, 964, 944, 885, 823, 789, 776, 687 cm^−1^: ^1^H NMR (CDCl_3_, 300 MHz): *δ* = 7.50 (bs, 1H, NH), 4.49 (dd, *J* = 9.6, 2.2 Hz, 2H, CH), 3.50–3.32 (m, 1H, 1 H of CH_2_NH), 3.13 (dddd, *J* = 15.5, 10.1, 5.7, 1.3 Hz, 1H, 1 H of CH_2_NH), 2.08 (dtt, *J* = 17.4, 11.0, 3.4 Hz, 3H, 3 H of CH_2_), 1.90–1.67 (m, 2H, 2 H of CH_2_), 1.66–1.49 (m, 1H, 1 H of CH_2_); ^13^C NMR (CDCl_3_, 75 MHz): *δ* = 170.1 (C=O), 159.5 (NC), 57.8 (CH), 41.7 (CH_2_), 31.3 (CH_2_), 28.5 (CH_2_), 26.8 (CH_2_); HRMS (ESI+) *m*/*z*: [M + Na]^+^ Calcd for C_7_H_10_N_2_NaO_2_^+^: 161.0685; Found: 161.0694.

**((4R,5R)-5-(1-acetoxy-2-oxo-2-(((S)-2-oxoazepan-3-yl)amino)ethyl)-2,2-dimethyl-1,3-dioxolan-4-yl)methyl acetate (16):** To a solution of DMSO (44 µL, 0.61 mmol), in dry CH_2_Cl_2_ (3 mL), at −70 °C, under a nitrogen atmosphere, a solution of oxalyl chloride in dry CH_2_Cl_2_ (2 M, 0.26 mL) was added. The solution was stirred for approximately 10 min, until effervescence ceased. A solution of **7a** (50 mg, 0.24 mmol) in dry CH_2_Cl_2_ (1 + 1 + 0.5 mL) was added dropwise, and the solution was stirred for 10 min at −70 °C. NEt_3_ (160 µL, 1.15 mmol) was then added, and the solution was stirred for 2 h at −50 °C. After this time, the reaction mixture was poured into a mixture of 5% aq (NH_4_)H_2_PO_4_ (5 mL) and 1 M HCl (0.1 mL) (final pH 4) and extracted with Et_2_O (20 + 10 mL). The organic layer was washed with brine (5 mL), dried (Na_2_SO_4_), and concentrated. The resulting crude aldehyde **15** was rapidly solubilized in CH_2_Cl_2_ (1 mL) under N_2_ and isocyanide **5** (37 mg, 0.27 mmol) and acetic acid (15 μL, 0.27 mmol) were added. After stirring for 7 h at room temperature, the solvent was removed and the residue was filtered on silica gel (PE/Acetone 3:2) to give **16** (68 mg, 69%) as a 20:80 (*syn*/*anti*) mixture of diastereoisomers (Colonna Hydro RP (2) 150 × 3 mm, 4 micron; flow = 0.5 mL/min; Vinj 5 µL; Temp: 26 °C Term. ON, VWD = 210 nm; MS: FullScan 100–800 *m*/*z* Positive, tic volt: 750V, Gradient A = H_2_O+0.1%FA C = MeOH+0.1% FA, 0 min A = 80%, 30 min A = 0%. R_t_ (*anti*) = 12.9 min, R_t_ (*syn*) = 13.3 min). **16 *anti*** and **16 *syn*** can be separated performing column chromatography on silica gel with PE/Acetone (from 1:1 to 3:2).

**16 *anti*:** amorphous solid; R_f_ = 0.14 (PE/Acetone 2:1); [α]_D_^25^ = +26.47 (*c* 0.85, CHCl_3_): ^1^H NMR (CDCl_3_, 300 MHz): *δ* = 7.57 (bd, *J* = 5.8 Hz, 1H, NHCH), 6.25 (bt, *J* = 6.4 Hz, 1H, NHCH_2_), 5.21 (d, *J* = 7.6 Hz, 1H, CHOAc), 4.57 (dd, *J* = 7.6, 5.8 Hz, 1H, CH-CHOAc), 4.54–4.43 (m, 2H, CH-CH_2_OAc and CHNH), 4.41 (dd, *J* = 11.4, 3.8 Hz, 2H, 1 H of CH_2_OAc), 4.11 (dd, *J* = 11.4, 6.8 Hz, 1H, 1 H of CH_2_OAc), 3.36–3.19 (m, 2H, CH_2_NH), 2.17 (s, 3H, OAc), 2.08 (s, 3H, OAc), 2.20–1.95 (m, 2H, 2 H of CH_2_), 1.90–1.74 (m, 2H, 2 H of CH_2_), 1.51 (s, 3H, CH_3_ acetonide), 1.56–1.30 (m, 2H, 2 H of CH_2_), 1.38 (s, 3H, CH_3_ acetonide); ^13^C NMR (CDCl_3_, 75 MHz): *δ* = 175.0 (C=O), 170.8 (C=O), 169.5 (C=O), 166.3 (C=O), 109.7 (Cq acetonide), 75.3 (CHNH), 75.3 (CH-CHOAc), 71.4 (CH-OAc), 62.4 (CH_2_OAc), 52.6 (CH-CH_2_OAc), 42.3 (CH_2_NH), 31.1 (CH_2_), 29.0 (CH_2_), 28.0 (CH_2_), 27.7 (CH_3_ acetonide), 25.3 (CH_3_ acetonide), 21.0 (CH_3_ of Ac), 20.8 (CH_3_ of Ac); HRMS (ESI+) *m*/*z*: [M + Na]^+^ Calcd for C_18_H_28_N_2_NaO_8_^+^: 423.1738; Found: 423.1736.

**16 *syn*:** colorless oil; R_f_ = 0.15 (PE/Acetone 2:1); ^1^H NMR (CDCl_3_, 300 MHz): *δ* = 7.46 (bd, *J* = 6.3 Hz, 1H, CHNH), 6.02 (bt, *J* = 6.4 Hz, 1H, CH_2_NH), 5.27 (d, *J* = 3.5 Hz, 1H, CHOAc), 4.62 (dd, *J* = 6.5, 3.5 Hz, 1H, CH-CHOAc), 4.53–4.39 (m, 2H, CHNH and CH-CH_2_OAc), 4.23 (dd, *J* = 11.4, 5.2 Hz, 1H, 1 H of CH_2_OAc), 4.15 (dd, *J* = 11.4, 6.9 Hz, 1H, 1 H of CH_2_OAc), 3.31–3.20 (m, 2H, CH_2_NH), 2.23 (s, 3H, OAc), 2.16–1.96 (m, 2H, CH_2_), 2.08 (s, 3H, OAc), 1.91–1.78 (m, 2H, CH_2_), 1.52 (s, 3H, CH_3_ acetonide), 1.48–1.37 (m, 2H, CH_2_), 1.35 (s, 3H, CH_3_ acetonide); ^13^C NMR (CDCl_3_, 75 MHz): δ = 174.9 (C=O), 170.8 (C=O), 169.7 (C=O), 166.8 (C=O), 109.7 (Cq acetonide), 75.8 (CH-CHOAc), 74.8 (CH-CH_2_OAc), 71.8 (CHOAc), 62.8 (CH_2_OAc), 52.4 (CHNH), 42.3 (CH_2_NH), 31.3 (CH_2_), 29.0 (CH_2_), 28.0 (CH_2_), 27.2 (OAc), 25.4 (OAc), 21.0 (CH_3_ acetonide), 20.9 (CH_3_ acetonide); HRMS (ESI+) *m*/*z*: [M + Na]^+^ Calcd for C_18_H_28_N_2_NaO_8_^+^: 423.1738; Found: 423.1736.

**2-((4S,5S)-5-((R,E)-1-((tert-butyldimethylsilyl)oxy)-4-methylpent-2-en-1-yl)-2,2-dimethyl-1,3-dioxolan-4-yl)-2-hydroxy-N-((S)-2-oxoazepan-3-yl)acetamide (3):** To a solution of DMSO (37 µL, 0.52 mmol), in dry CH_2_Cl_2_ (3 mL), at −70 °C, under a nitrogen atmosphere, a solution of oxalyl chloride in dry CH_2_Cl_2_ (1.43 M, 0.33 mL) was added. The solution was stirred for approximately 10 min, until effervescence ceased. A solution of 12 (67 mg, 0.19 mmol) in dry CH_2_Cl_2_ (1 + 0.5 mL) was added dropwise, and the solution was stirred for 10 min at −70 °C. NEt_3_ (160 µL, 1.15 mmol) was then added, and the solution was stirred for 2 h at −50 °C. After this time, the reaction mixture was poured into a mixture of 5% aq (NH_4_)H_2_PO_4_ (5 mL) and 1 M HCl (0.1 mL) (final pH 4) and extracted with Et_2_O (20 + 10 mL). The organic layer was washed with brine (5 mL), dried (Na_2_SO_4_), and concentrated. The resulting crude aldehyde **4** was rapidly solubilized in *i*Pr_2_O (500 µL) under N_2,_ and isocyanide **5** (53 mg, 0.38 mmol) and acetic acid (22 μL, 0.38 mmol) were added. After stirring for 48 h at room temperature, the solvent was removed and the residue was filtered on silica gel (PE/AcOEt 3:4) to give a mixture of products **18** and **3** (73 mg), which was treated with MeOH/H_2_O/Et_3_N (5:1:1) and stirred at room temperature for 48 h. Then, the solvent was removed. The diastereomeric ratio was determined as 80:20 (*anti*:*syn*) using reverse-phase HPLC on the crude mixture (C6 PHENYLIC RP column (150 × 3 mm, 3 μ) at 30 °C with flow = 0.34 mL/min and gradient H_2_O/MeOH, A = MeOH + 0.1% FA—B = H_2_O + 0.1% FA, 0 min B = 30%, 20 min B = 20%. Detection was carried out with UV at 210 nm, R_t_ (*anti*) = 13.2 min, R_t_ (*syn*) = 15.6 min). The crude residue was purified with column chromatography of silica gel (PE/Et_2_O 1:20) to give **3 *anti*** (55 mg, 58%) and **3 *syn*** (14 mg, 16%). **3 *anti***: pale-yellow oil, R_f_ = 0.35 (Et_2_O/PE 20:1); [α]_D_^20^ = +16.9 (*c* 1.0, CHCl_3_); IR (ATR): ν = 3462, 3346, 2956, 2931, 2858, 1685, 1598, 1525, 1463, 1375, 1254, 1215, 1168, 1069, 1045, 974, 834, 800, 777, 667 cm^−1^; ^1^H NMR (CDCl_3_, 300 MHz): *δ* = 7.79 (bd, *J* = 6.5 Hz, 1H, NHCH), 6.20 (bs, 1H, NHCH_2_), 5.63 (dd, *J* = 15.5, 6.6 Hz, 1H, *i*Pr-CH=CH), 5.47 (dd, *J* = 15.5, 7.8 Hz, 1H, *i*Pr-CH=CH), 4.64–4.53 (m, 3H, OH, NHCH and CH-OTBS), 4.30–4.19 (m, 2H, CHOH and CH-CHOH), 4.06 (t, *J* = 4.5 Hz, 1H, CH-CHOTBS), 3.38–3.15 (m, 2H, CH_2_NH), 2.31 (h, *J* = 6.6 Hz, 1H, CH of *i*Pr), 2.21–2.06 (m, 1H, 1 H of CH_2_), 2.07–1.94 (m, 1H, 1 H of CH_2_), 1.93–1.70 (m, 2H, 2 H of CH_2_), 1.62–1.47 (m, 1H, 1 H of CH_2_), 1.52 (s, 3H, CH_3_ acetonide), 1.47–1.36 (m, 1H, 1 H of CH_2_), 1.33 (s, 3H, CH_3_ acetonide), 1.005 (d, *J* = 6.7 Hz, 3H, CH_3_ of *i*Pr), 1.00 (d, *J* = 6.8 Hz, 3H, CH_3_ of *i*Pr), 0.90 (s, 9H, 3 CH_3_ of TBS), 0.14 (s, 3H, CH_3_ of TBS), 0.11 (s, 3H, CH_3_ of TBS); ^13^C NMR (CDCl_3_, 75 MHz): *δ* = 175.5 (C=O), 170.6 (C=O), 142.0 (*i*Pr-CH=CH), 125.9 (*i*Pr-CH=CH), 108.5 (Cq acetonide), 80.8 (CH-CHOTBS), 77.9 (CHOH), 73.6 (CHOTBS), 70.2 (CH-CHOH), 52.2 (CHNH), 42.1 (CH_2_NH), 31.4 (CH_2_), 30.8 (CH of *i*Pr), 29.0 (CH_2_), 28.0 (CH_2_), 27.7 (CH_3_ acetonide), 25.9 (3 CH_3_ of TBS), 25.6 (CH_3_ acetonide), 22.1 (CH_3_ of *i*Pr), 21.9 (CH_3_ of *i*Pr), 18.3 (Cq of TBS), −3.9 (CH_3_ of TBS), −4.3 (CH_3_ of TBS); HRMS (ESI+) *m*/*z*: [M + Na]^+^ Calcd for C_25_H_46_N_2_NaO_6_Si^+^: 521.3017; Found: 521.3018. **3 *syn***: pale-yellow foam R_f_ = 0.25 ((Et_2_O/PE 20:1); [α]_D_^20^ = −33.7 (*c* 1.2, CHCl_3_); IR (ATR): ν = 3462, 3346, 2956, 2931, 2858, 1685, 1598, 1525, 1463, 1375, 1254, 1215, 1168, 1069, 1045, 974, 834, 800, 777, 667 cm^−1^; ^1^H NMR (CDCl_3_, 300 MHz): *δ* = 7.98 (d, *J* = 7.0 Hz, 1H, NHCH), 6.16 (bs, 1H, NHCH_2_), 5.72 (dd, *J* = 15.6, 6.9 Hz, 1H, *i*Pr-CH=CH), 5.38 (dd, *J* = 15.5, 6.1 Hz, 1H, *i*Pr-CH=CH), 4.68 (t, *J* = 4.9 Hz, 1H, CH-OTBS), 4.64–4.56 (m, 2H, NHCH and CH-CHOH), 4.54 (d, *J* = 2.6 Hz, 1H, OH), 4.29 (d, *J* = 2.2 Hz, 1H, CHOH), 4.17 (dd, *J* = 6.5, 4.4 Hz, 1H, CH-CHOTBS), 3.40–3.13 (m, 2H, CH_2_NH), 2.31 (h, *J* = 6.9 Hz, 1H, CH of *i*Pr), 2.13–1.92 (m, 2H, 2 H of CH_2_), 1.92–1.64 (m, 2H, 2 H of CH_2_), 1.61–1.51 (m, 1H, 1 H of CH_2_), 1.49 (s, 3H, CH_3_ acetonide), 1.32 (s, 3H, CH_3_ acetonide), 0.99 (d, *J* = 6.7 Hz, 6H, 2 CH_3_ of *i*Pr), 0.92 (s, 9H, 3 CH_3_ of TBS), 0.14 (s, 3H, CH_3_ of *i*Pr), 0.11 (s, 3H, CH_3_ of *i*Pr); ^13^C NMR (CDCl_3_, 75 MHz): *δ* = 175.6 (C=O), 170.8 (C=O), 141.2 (*i*Pr-CH=CH), 124.6 (*i*Pr-CH=CH), 108.3 (Cq acetonide), 79.2 (CH-CHOTBS), 77.8 (CH-CHOH), 72.5 (CHOTBS), 71.5 (CHOH), 51.9 (CHNH), 42.3 (CH_2_NH), 31.9 (CH_2_), 31.0 (CH of *i*Pr), 29.2 (CH_2_), 28.2 (CH_2_), 26.4 (CH_3_ acetonide), 26.0 (3 CH_3_ of *i*Pr), 25.5 (CH_3_ acetonide), 22.3 (CH_3_ of *i*Pr), 22.2 (CH_3_ of *i*Pr), 18.6 (Cq TBS), −4.2 (CH_3_ of TBS), −4.7 (CH_3_ of TBS); HRMS (ESI+) *m*/*z*: [M + Na]^+^ Calcd for C_25_H_46_N_2_NaO_6_Si^+^: 521.3017; Found: 521.3018.

**(R)-2-((4R,5S)-5-((R,E)-1-((tert-butyldimethylsilyl)oxy)-4-methylpent-2-en-1-yl)-2,2-dimethyl-1,3-dioxolan-4-yl)-2-methoxy-N-((S)-2-oxoazepan-3-yl)acetamide (19):** A solution of **3 *anti*** (64 mg, 0.128 mmol) in dry THF (1 mL) under a N_2_ atmosphere was cooled at −10 °C. NaH (60% in silicon oil, 8 mg, 0.199 mmol) was added and the mixture was stirred for 30 min. Then, MeI (17 µL, 0.265 mmol) was added and the reaction was stirred at −10 °C for 48 h. The reaction mixture was diluted with saturated NH_4_Cl solution, extracted with CH_2_Cl_2_, dried (Na_2_SO_4_), and concentrated. The crude residue was purified using silica gel column chromatography (PE/Et_2_O 1:20) to give **20** (14 mg, 14%) and **19** (59 mg, 66%) both as a colorless oil. **19**: R_f_ = 0.31 (Et_2_O + 2% AcOEt); [α]_D_^20^ = +6.2 (*c* 1.7, CHCl_3_); IR (ATR): ν = 3383, 3292, 2955, 2930, 2858, 1714, 1662, 1504, 1474, 1435, 1362, 1334, 1250, 1217, 1169, 1103, 1073, 1047, 1017, 972, 941, 899, 875, 834, 808, 776, 754, 666 cm^−1^; ^1^H NMR (CDCl_3_, 300 MHz): *δ* = 7.53 (d, *J* = 6.2 Hz, 1H, NHCH), 6.02 (t, *J* = 6.6 Hz, 1H, NHCH_2_), 5.66 (dd, *J* = 15.6, 5.8 Hz, 1H, *i*Pr-CH=CH), 5.56 (dd, *J* = 15.6, 6.7 Hz, 1H, *i*Pr-CH=CH), 4.75–4.52 (m, 2H, CHOTBS and NHCH), 4.22 (dd, *J* = 7.1, 6.2 Hz, 1H, CH-CHOCH_3_), 4.09 (dd, *J* = 6.1, 4.6 Hz, 1H, CH-CHOTBS), 4.05 (d, *J* = 7.0 Hz, 1H, CHOCH_3_), 3.34 (s, 3H, OCH_3_), 3.32–3.20 (m, 2H, CH_2_NH), 2.32 (h, *J* = 6.1 Hz, 1H, CH of *i*Pr), 2.22–2.08 (m, 1H, 1 H of CH_2_), 2.04–1.94 (m, 1H, 1 H of CH_2_), 1.91–1.78 (m, 2H, 2 H of CH_2_), 1.60–1.40 (m, 2H, 2 H of CH_2_), 1.41 (s, 3H, CH_3_ acetonide), 1.29 (s, 3H, CH_3_ acetonide), 1.02 (d, *J* = 6.7 Hz, 3H, CH_3_ of *i*Pr), 1.00 (d, *J* = 6.7 Hz, 3H, CH_3_ of *i*Pr), 0.90 (s, 9H, 3 CH_3_ of TBS), 0.10 (s, 3H, CH_3_ of TBS), 0.07 (s, 3H, CH_3_ of TBS); ^13^C NMR (CDCl_3_, 75 MHz): *δ* = 175.3 (C=O), 169.7 (C=O), 140.9 (*i*Pr-CH=CH), 127.3 (*i*Pr-CH=CH), 108.5 (Cq acetonide), 81.0 (CH-CHOTBS), 80.9 (CHOCH_3_), 77.6 (CH-CHOCH_3_), 73.3 (CHOTBS), 57.4 (OCH_3_), 52.0 (CHNH), 42.2 (NHCH_2_), 31.5 (CH_2_), 30.9 (CH of *i*Pr), 29.1 (CH_2_), 28.0 (CH_2_), 27.2 (CH_3_ acetonide), 26.1 (3 CH_3_ of TBS), 25.3 (CH_3_ acetonide), 22.4 (CH_3_ of *i*Pr), 22.0 (CH_3_ of *i*Pr), 18.5 (Cq of TBS), −3.5 (CH_3_ of TBS), −4.3 (CH_3_ of TBS); HRMS (ESI+) *m*/*z*: [M + Na]^+^ Calcd for C_25_H_46_N_2_NaO_6_Si^+^: 535.3174; Found: 535.3177.

**(R)-2-((4R,5S)-5-((R,E)-1-((tert-butyldimethylsilyl)oxy)-4-methylpent-2-en-1-yl)-2,2-dimethyl-1,3-dioxolan-4-yl)-2-methoxy-N-((S)-1-methyl-2-oxoazepan-3-yl)acetamide (20):** colorless oil; R_f_ = 0.38 (Et_2_O + 2% AcOEt); [α]_D_^20^ = +1.6 (*c* 0.6, CHCl_3_); IR (ATR): ν = 3388, 2955, 2930, 2858, 2246, 1648, 1495, 1461, 1403, 1381, 1370, 1339, 1251, 1214, 1157, 1138, 1102, 1075, 1047, 1016, 973, 910, 879, 834, 808, 777, 729, 646 cm^−1^; ^1^H NMR (CDCl_3_, 300 MHz): *δ* = 7.61 (bd, *J* = 6.0 Hz, 1H, NHCH), 5.78–5.46 (m, 2H, CH=CH), 4.70 (dd, *J* = 9.5, 6.3 Hz, 1H, NHCH), 4.56 (dd, *J* = 6.4, 4.3 Hz, 1H, CHOTBS), 4.20 (dd, *J* = 7.4, 6.2 Hz, 1H, CH-CHOMe), 4.08 (dd, *J* = 6.0, 4.2 Hz, 1H, CH-CHOTBS), 4.02 (d, *J* = 7.5 Hz, 1H, CHOMe), 3.61 (dd, *J* = 15.3, 11.6 Hz, 1H, 1 H of CH_2_-N), 3.33 (s, 3H, OMe), 3.18 (dd, *J* = 15.0, 4.5 Hz, 1H, 1 H of CH_2_-N), 3.04 (s, 3H, NMe), 2.31 (h, *J* = 6.0 Hz, 1H, CH of *i*Pr), 2.21–2.00 (m, 1H, 1 H of CH_2_), 2.00–1.72 (m, 3H, 3 H of CH_2_), 1.55–1.31 (m, 2H, 2 H of CH_2_), 1.42 (s, 3H, CH_3_ acetonide), 1.29 (s, 3H, CH_3_ acetonide), 1.01 (d, *J* = 6.7 Hz, 3H, CH_3_ of *i*Pr), 1.00 (d, *J* = 6.7 Hz, 3H, CH_3_ of *i*Pr), 0.90 (s, 9H, 3 CH_3_ of TBS), 0.10 (s, 3H, CH_3_ of TBS), 0.07 (s, 3H, CH_3_ of TBS); ^13^C NMR (CDCl_3_, 75 MHz): *δ* = 172.8 (C=O), 169.5 (C=O), 140.7 (CH=), 127.3 (C=H), 108.5 (Cq acetonide), 81.2 (CH-CHOTBS), 80.8 (CHOMe), 77.5 (CH-CHOMe), 73.3 (CHOTBS), 57.3 (OMe), 51.9 (NHCH), 50.5 (CH_2_N), 36.0 (NMe), 31.7 (CH_2_), 30.9 (CH of *i*Pr), 27.8 (CH_2_), 27.3 (CH_3_ acetonide), 26.8 (CH_2_), 26.2 (3 CH_3_ of TBS), 25.3 (CH_3_ acetonide), 22.4 (CH_3_ of *i*Pr), 22.1 (CH_3_ of *i*Pr), 18.5 (Cq of TBS), −3.6 (CH_3_ of TBS), −4.3 (CH_3_ of TBS); HRMS (ESI+) *m*/*z*: [M + Na]^+^ Calcd for C_27_H_50_N_2_NaO_6_Si^+^: 549.3330; found: 549.3299.

**(2R,3R,4R,5R,E)-3,4,5-trihydroxy-2-methoxy-8-methyl-N-((S)-2-oxoazepan-3-yl)non-6-enamide (2):** Pale-yellow foam; R_f_ = 0.42 (AcOEt + 2% MeOH); [α]_D_^20^ = +34.5 (*c* 0.3, CHCl_3_); IR (ATR): ν = 3334, 2956, 2930, 2868, 1639, 1519, 1483, 1437, 1361, 1334, 1291, 1261, 1066, 973, 943, 893, 800, 720 cm^−1^; ^1^H NMR (CDCl_3_, 300 MHz): *δ* = 7.63 (bd, *J* = 7.0 Hz, 1H, NHCH), 6.22 (bt, *J* = 6.1 Hz, 1H, NHCH_2_), 5.78 (ddd, *J* = 15.6, 6.4, 0.9 Hz, 1H, *i*Pr-CH=CH), 5.55 (ddd, *J* = 15.6, 7.2, 1.3 Hz, 1H, *i*Pr-CH=CH), 4.59 (dd, *J* = 10.3, 7.0 Hz, 1H, NHCH), 4.26 (bt, *J* = 5.3 Hz, 1H, CH-5), 4.07 (d, *J* = 3.3 Hz, 1H, CH-2), 4.07–3.95 (m, 1H, CH-3), 3.77 (d, *J* = 4.7 Hz, 1H, OH), 3.62 (dt, *J* = 8.2, 4.4 Hz, 1H, CH-4), 3.49 (s, 3H, OMe), 3.49–3.47 (m, 1H, OH), 3.39–3.18 (m, 2H, NHCH_2_), 2.77 (bs, 1H, OH), 2.32 (h, *J* = 6.3 Hz, 1H, CH of *i*Pr), 2.11–2.00 (m, 2H, 2 H of CH_2_), 1.94–1.71 (m, 2H, 2 H of CH_2_), 1.61–1.36 (m, 2H, 2 H of CH_2_), 1.01 (d, *J* = 6.8 Hz, 6H, 2 CH_3_ of *i*Pr); ^13^C NMR (CDCl_3_, 75 MHz): *δ* = 175.2 (C=O), 170.7 (C=O), 141.8 (*i*Pr-CH=CH), 125.3 (*i*Pr-CH=CH), 82.6 (CH-2), 74.7 (CH-5), 74.0 (CH-4), 73.0 (CH-3), 58.9 (OMe), 52.5 (NHCH), 42.2 (NHCH_2_), 31.0 (CH of *i*Pr), 30.8 (CH_2_), 28.9 (CH_2_), 28.1 (CH_2_), 22.5 (CH_3_ of *i*Pr), 22.3 (CH_3_ of *i*Pr); HRMS (ESI+) *m*/*z*: [M + Na]^+^ Calcd for C_17_H_30_N_2_NaO_6_^+^: 381.1996; Found: 381.1993.

**(3aR,4R,7R,7aR)-7-methoxy-2,2-dimethyl-4-((E)-3-methylbut-1-en-1-yl)tetrahydro-6H-[1,3]dioxolo[4,5-c]pyran-6-one (21):** Colorless oil; R_f_ = 0.69 (AcOEt + 10% MeOH); ^1^H NMR (CDCl_3_, 300 MHz): *δ* = 5.82 (ddd, *J* = 15.8, 6.8, 2.1 Hz, 1H, *i*Pr-CH=CH), 5.45 (ddd, *J* = 15.9, 4.1, 1.3 Hz, 1H, *i*Pr-CH=CH), 5.07–4.98 (m, 1H, CH-5), 4.82 (dd, *J* = 7.6, 3.5 Hz, 1H, CH-3), 4.56 (dd, *J* = 7.6, 1.1 Hz, 1H, CH-4), 4.07 (d, *J* = 3.5 Hz, 1H, CH-2), 3.63 (s, 3H, OMe), 2.37 (sest, *J* = 6.8 Hz, 1H, CH of *i*Pr), 1.50 (s, 3H, CH_3_ acetonide), 1.36 (s, 3H, CH_3_ acetonide), 1.02 (d, *J* = 6.7 Hz, 6H, 2 CH_3_ of *i*Pr); ^13^C NMR (CDCl_3_, 75 MHz): *δ* = 168.6 (C=O), 142.7 (*i*Pr-CH=CH), 121.3 (*i*Pr-CH=CH), 111.0 (Cq acetonide), 79.7 (CH-5), 76.2 (CH-2), 75.7 (CH-4), 74.8 (CH-3), 59.9 (OMe), 31.3 (CH of *i*Pr), 26.2 (CH_3_ acetonide), 24.4 (CH_3_ acetonide), 22.1 (CH_3_ of *i*Pr), 22.0 (CH_3_ of *i*Pr); HRMS (ESI+) *m*/*z*: [M + Na]^+^ Calcd for C_14_H_22_NaO_5_^+^: 293.1359; Found: 293.1349.

## 4. Conclusions

In conclusion, we have reported herein the first total synthesis of 4-*epi*-Bengamide E with a 4.1% yield over 14 steps from D-isoascorbic acid (with a 6.6% yield over 12 steps from *meso*-diol **1**). Key features of our convergent synthesis included (a) a chemoenzimatic desymmetrization of a biobased achiral compound; (b) a nucleophilic addition of acetylides and subsequent selective reduction to form the *E-*allylic alcohol; and (c) a diastereoselective Passerini reaction. This strategy allows the easy variation of side-chains, using different alkynes or different isocyanoamides in the synthetic sequence. Regarding concerns for stereochemical diversity, we think that there will be future chances to explore it. For example, configuration at C-5 might be varied by developing a complementary *syn*-selective reduction of ketoalkyne. Configuration at C-2 can be inverted by performing a Mitsunobu inversion before methylation. Finally, it must be noted that the enantiomeric series is easily accessible, due to the availability of the enantiomer of **7b**. The main limitation of this approach, from the stereochemical point of view, is the fixed relative configuration between C-3 and C-4, which prevents access to Bengamide E itself. For the Bengamide E relative configuration, tartaric acid-derived building blocks are surely better suited, which is already demonstrated by some total syntheses. These synthetic efforts, as well as biological evaluations of the described bengamide analogues, are currently in progress and will be presented in due course. 

## Data Availability

Data are contained within the article and Appendix A.

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
