# Peer review of "Total Synthesis of 4-epi-Bengamide E"

_molecules, 2024, doi:10.3390/molecules29081715_

Round 1

Reviewer 1 Report

Comments and Suggestions for Authors

Comment- molecules-2951430

Recommend Acceptance: Published after minor revision

The presented work on Bengamides is a significant contribution to the field of organic synthesis and natural product chemistry. In this study, Moni and colleagues effectively highlight the importance of Bengamides, particularly Bengamide E, due to their remarkable anticancer activity and unique structural features. The introduction of a highly convergent, multi-component reaction (MCR)-based approach for the synthesis of 4-epi-Bengamide E represents a notable advancement, allowing for the exploration of stereochemical diversity and the potential for preparing diverse analogs efficiently.

The utilization of a diastereoselective Passerini reaction with a chiral, enantiomerically pure aldehyde is a particularly innovative aspect of the methodology. This approach not only offers atom and step economy but also demonstrates the sustainability and flexibility of MCR-based strategies in natural product synthesis. Moreover, the authors' emphasis on the use of enantiopure chiral building blocks synthesized from biobased starting materials showcases a thoughtful integration of green chemistry principles into the synthetic route.

Overall, the methodology presented in the paper is logically structured and well-supported by evidence from the literature and the authors' own experimental work. The conclusions drawn are consistent with the arguments presented, reflecting a deep understanding of both the synthetic challenges and the biological significance of the target compounds. Additionally, the references provided are appropriately chosen and contribute to the credibility of the study by acknowledging prior research in the field. Considering the novelty and the importance of this work, we recommend the acceptance of this work to molecules for publication after addressing the following minor points.

1.            The correct term is "Scheme 6" in line 154 of the article.

2.            To avoid confusion, it's recommended to use letters instead of numbers to label the table and annotations.

3.  The authors should label the specific positions of the C1-C2 and C5-C6 bonds in Scheme 1 and Scheme 2.

4.  Since the Passerini reaction is the pivotal step in this synthetic plan, it would be beneficial for the authors to present the corresponding mechanism in the text to facilitate readers' understanding.

Author Response

Reviewer: 1

  1. The presented work on Bengamides is a significant contribution to the field of organic synthesis and natural product chemistry. In this study, Moni and colleagues effectively highlight the importance of Bengamides, particularly Bengamide E, due to their remarkable anticancer activity and unique structural features. The introduction of a highly convergent, multi-component reaction (MCR)-based approach for the synthesis of 4-epi-Bengamide E represents a notable advancement, allowing for the exploration of stereochemical diversity and the potential for preparing diverse analogs efficiently.

The utilization of a diastereoselective Passerini reaction with a chiral, enantiomerically pure aldehyde is a particularly innovative aspect of the methodology. This approach not only offers atom and step economy but also demonstrates the sustainability and flexibility of MCR-based strategies in natural product synthesis. Moreover, the authors' emphasis on the use of enantiopure chiral building blocks synthesized from biobased starting materials showcases a thoughtful integration of green chemistry principles into the synthetic route.

Overall, the methodology presented in the paper is logically structured and well-supported by evidence from the literature and the authors' own experimental work. The conclusions drawn are consistent with the arguments presented, reflecting a deep understanding of both the synthetic challenges and the biological significance of the target compounds. Additionally, the references provided are appropriately chosen and contribute to the credibility of the study by acknowledging prior research in the field. Considering the novelty and the importance of this work, we recommend the acceptance of this work to molecules for publication after addressing the following minor points.

We thank the reviewer for the positive feedback on our manuscript.

  1. The correct term is "Scheme 6" in line 154 of the article.

We thank the referee for noticing. Mistake has been corrected throughout the manuscript.

  1. To avoid confusion, it's recommended to use letters instead of numbers to label the table and annotations.

We agree with the referee. We replaced the numbers with the letters in Tables 1 and 2, and we transfer the comment from reference 43 to the main text, as suggested by the referee 2 too.

  1. The authors should label the specific positions of the C1-C2 and C5-C6 bonds in Scheme 1 and Scheme 2.

We agree with the referee. We highlighted the bonds in the Scheme 1 and 2.

  1. Since the Passerini reaction is the pivotal step in this synthetic plan, it would be beneficial for the authors to present the corresponding mechanism in the text to facilitate readers' understanding.

We added a sentence regarding Passerini reaction and a Scheme with the mechanism in the manuscript.

Reviewer 2 Report

Comments and Suggestions for Authors

The manuscript "Total Synthesis of 4-epi-Bengamide E," presented by Lisa Moni et al., summarizes the importance of the family of compounds known as Bengamides, from their isolation from Jaspidea sponges to the biological activity exhibited by Bengamide E and some of its described epimers. The authors note that 4-epi-Bengamide E had not been synthesized to date, and therefore, it is unknown if the biological activity of Bengamide E is observed in this new compound. This manuscript leaves us with the question of whether 4-epi-Bengamide E is biologically active or not. Likewise, if understanding its biological activity is not a central objective of this research, why consider synthesizing an epimer of a natural compound? The norm is to aim for the natural product and not its isomers.

Personally, this manuscript encapsulates everything interesting and challenging about the organic synthesis of natural products. It is well-presented and describes the process in detail, meeting the necessary criteria for publication in Molecules. However, there are some points that need to be addressed:

1- It is necessary to evaluate 4-epi-Bengamide E's antitumor properties, especially in cell lines where Bengamide E is active. Additionally, the antibiotic properties could be evaluated in strains where Bengamide E is active. With at least one of these analyses, and by comparing with the natural compound, this work will demonstrate its full potential.

2- The conclusion must be modified. The authors declare a 6.6% overall yield from meso-diol 1 (12 stages), but they actually start their synthesis from D-isoascorbic acid and/or meso-erythritol, which adds synthesis stages to the real synthetic route.

3- Unfortunately, there are two points in the route where the loss of diastereoselectivity impacts the overall performance of the synthesis, such as the transition from 9 syn/anti to 9 syn + 9 anti passing through the intermediate propargyl ketone. I suggest two reactions that could be especially useful in this case: the Corey-Itsuno reduction (J. Am. Chem. Soc. 1996, 118, 10938) and the transfer hydrogenation catalyzed by Ru(II) of Noyori (J Am. Chem. Soc. 1997, 119, 8738). Between these two reactions, I recommend the Noyori reduction due to its simplicity and the ease of obtaining the catalyst. If possible, this reaction should be carried out.

4- Transfer the comment from reference 43 to the main text. The Molecules format is only for references and not for notes.

Author Response

Reviewer: 2

  1. The manuscript "Total Synthesis of 4-epi-Bengamide E," presented by Lisa Moni et al., summarizes the importance of the family of compounds known as Bengamides, from their isolation from Jaspidea sponges to the biological activity exhibited by Bengamide E and some of its described epimers. The authors note that 4-epi-Bengamide E had not been synthesized to date, and therefore, it is unknown if the biological activity of Bengamide E is observed in this new compound. This manuscript leaves us with the question of whether 4-epi-Bengamide E is biologically active or not. Likewise, if understanding its biological activity is not a central objective of this research, why consider synthesizing an epimer of a natural compound? The norm is to aim for the natural product and not its isomers.

Personally, this manuscript encapsulates everything interesting and challenging about the organic synthesis of natural products. It is well-presented and describes the process in detail, meeting the necessary criteria for publication in Molecules. However, there are some points that need to be addressed:

Answer 1: we thank the reviewer for the feedback on our manuscript.

  1. It is necessary to evaluate 4-epi-Bengamide E's antitumor properties, especially in cell lines where Bengamide E is active. Additionally, the antibiotic properties could be evaluated in strains where Bengamide E is active. With at least one of these analyses, and by comparing with the natural compound, this work will demonstrate its full potential.

We thank the referee for the suggestion. Regarding the biological assays to evaluate antitumor activity in cell and antibiotic activity, we don’t have this expertise in our group. However, we have planned to carry out the biological assays in collaboration with some colleagues once we will obtain a set of analogues by applying our synthetic strategy. As highlighted in the introduction and the conclusions of the manuscript, this work focused on the development of a convergent synthesis, which allows the easy introduction of diverse side-chains and the exploration of stereochemical diversity. The synthesis of the unprecedented isomer of natural Bengamide E is just a first example. For this reason, these experiments will be the subject of a subsequent publication.

  1. The conclusion must be modified. The authors declare a 6.6% overall yield from meso-diol 1 (12 stages), but they actually start their synthesis from D-isoascorbic acid and/or meso-erythritol, which adds synthesis stages to the real synthetic route.

We agree with the referee. We added this sentence in the conclusion: “in 4.1% yield over 14 steps from D-isoascorbic acid”.

  1. Unfortunately, there are two points in the route where the loss of diastereoselectivity impacts the overall performance of the synthesis, such as the transition from 9 syn/anti to 9 syn + 9 anti passing through the intermediate propargyl ketone. I suggest two reactions that could be especially useful in this case: the Corey-Itsuno reduction (J. Am. Chem. Soc. 1996, 118, 10938) and the transfer hydrogenation catalyzed by Ru(II) of Noyori (J Am. Chem. Soc. 1997, 119, 8738). Between these two reactions, I recommend the Noyori reduction due to its simplicity and the ease of obtaining the catalyst. If possible, this reaction should be carried out.

We thank the referee for the suggestions. We are aware that the diastereomeric ratio obtained in the reduction of propargyl ketone is not high. However, we decided to especially study the diastereoselective reduction, and we chose not to use chiral catalysts or chiral stoichiometric reducing agents (i.e. Alpine borane®). This would have complicated the synthesis, introducing matched/mismatched issues on a quite complex substrate. Anyway, this possibility will be explored in the further study for the selective obtainment of syn and anti isomers for the preparation of the different stereoisomers at C-5.

  1. Transfer the comment from reference 43 to the main text. The Molecules format is only for references and not for notes.

Answer 5: we transfer the comment from reference 43 to the main text, as suggested by the referee